# Acceptability determinants of a proposal to reduce antiretroviral treatment to an oral two-drug regimen among patients living with HIV and physicians in France

Anne-Sophie Petit[1]*, Clotilde Allavena[2], David Zucman[3], Laurent Hocqueloux[4], Olivia Rousset-Torrente[1], Guillaume Roucoux[1], Claudine Duvivier[5], Gwenaël Le Moal[6], Olivier Chassany[1], Martin Duracinsky[1]

1 URC-ECO, Hotel-Dieu Hospital, AP-HP, Paris, France, 2 Department of Infectious and Tropical Diseases, University Hospital, Nantes, France, 3 Department of Internal Medicine, Foch Hospital, Suresnes, France, 4 Department of Internal Medicine, Regional Hospital Centre, Orléans, France, 5 Department of Infectious and Tropical Diseases, Necker Hospital, AP-HP, Paris, France, 6 Department of Infectious and Tropical Diseases, CHU de Poitiers, Poitiers, France

* anne-sophie.petit-ext@aphp.fr

**Data Availability Statement:** All relevant data are within the manuscript and its Supporting Informations files.

## Abstract

An oral two-drug regimen (O2DR) in the form of a once-a-day single tablet is now recommended for treatment switching and treatment initiation for HIV. In clinical care, the process of treatment change refers to adaptation issues, both individual and within the care relationship. The study aim is to present the determinants involved in the acceptability of switching to O2DR in the PROBI (Patient-Reported Outcomes BItherapy) qualitative study. The study includes 30 interviews: 15 were conducted with doctors caring for people living with HIV, 15 were conducted with patients who had been offered a change of treatment. A double analysis was carried out: lexicometric analysis to highlight the structuring of the discourse around the change in treatment and a thematic analysis to understand the associated issues more precisely. The results highlighted common concerns with respect to switching to O2DR. Also, the caregiver-patient relationship was a central determinant in treatment switching. Information, knowledge and representations of O2DR are also factors facilitating treatment change and should be taken into account for doctors' and patients' adherence.

## Introduction

An oral two-drug regimen (O2DR) in the form of a once-a-day single tablet is now widely recommended for both switching and initiating treatment for people living with HIV (PLWH) [1]. The efficacy and safety of dual therapy switch in terms of resistance and viral load maintenance have been proven in previous studies [2–6]. Since 2019, two oral O2DR have been approved by health authorities, and this treatment is seen as a significant therapeutic advance and an individual adaptation strategy for PLWH [7, 8]. Despite the established clinical efficacy of O2DR, to our knowledge, there have been few studies that have specifically investigated the

**Funding:** Funded studies None of the authors have received award, honoraria from the funder. The funder is Viiv Healthcare in program of investigational sponsor research. The URL of funder website is: https://viiv-portal.idea-point. com/Investigator-Sponsored-Study.aspx The funders had no role in study design, date collection and analysis, decision to publish, or preparation of the manuscript.

**Competing interests:** I have read the journal's Policy and the authors of this manuscript have the following competing interests: CD has received consultancies, speaker honoraria and travel grants from Gilead, Janssen-Cilag, MSD and Viiv Healthcare; GM has received fees for participation in advisory boards or presentations from Gilead and MSD; CA has received travel grants and honoraria from Gilead, Viiv Healthcare and MSD; LH has received fees for participation in advisory boards or présentations from Viiv Healthcare and Merck, LH has received travel support for congress attendance from Viiv Healthcare, Merck, Gilead and Shape and Dohme; MD has received consultancies, speaker honoraria and travel grants from Gilead, MSD and Viiv Healthcare; OC has received consultancies and speaker honoraria from Gilead and Viiv Healthcare. The following authors have declared that no competing interests exist: ASP, OT, GR and DZ.

psychosocial determinants at stake in the context of treatment change. Literature on treatment adherence has shown that patients' adherence to antiretroviral therapy is influenced by various factors, such as their beliefs about the medication, the complexity of the regimen, and their social support [9, 10]. Shared decision making, which involves patient participation in treatment decisions, has been associated with improved adherence and treatment outcomes [11, 12]. Understanding these factors and their impact on patient perception of treatment, as well as quality of life, is crucial for individualizing and optimizing HIV treatment regimens. In clinical care, the process of treatment change refers to adaptation issues, both individual and within the care relationship. Thus, patient-reported outcomes are crucial in understanding treatment perception and, more globally, in understanding quality of life issues [13] to adapt and individualize treatment. The PROBI mixed-methods study aims to assess acceptability, perceived toxicity, preferences, and quality of life issues in patients who have undergone a treatment change to O2DR. Also, the switch to O2DR has been approved by health authorities from Europe and North American. The study consists of both qualitative and quantitative phases. The qualitative PROBI phase serves as a preliminary investigation that meets two objectives: firstly, to investigate the knowledge, beliefs, and representations of O2DR among patients and physicians, and secondly, to create specific items within the PROQOL-HIV scale to measure the quality of life of patients who have switched to O2DR. The objective of this article is to present the determinants involved in the acceptability of switching to O2DR in the PROBI qualitative phase.

# Method

## Samples

The qualitative phase interviewed adult PLWH and physicians. For both populations, the sampling was based on case identification [14] followed by a theoretical sampling logic, incorporating criteria sought as indicators of the phenomenon in question. In order to obtain a diverse range of perspectives and practices regarding treatment change, the recruitment of physicians was conducted in 11 different hospital centers. This approach allowed for a greater variety of experiences and opinions to be represented in the study. Similarly, participants living with HIV were recruited from 7 different hospital centers to ensure a wide range of experiences and perspectives. This strategy was important to capture the complexity and diversity of the population of interest, and to ensure that the findings of the study are applicable across different settings. The inclusion of participants from multiple centers also helped to mitigate potential biases that may have arisen from recruiting participants from a single site. The study included adult PLWH who were fluent in French and currently receiving antiretroviral treatment. Participants were selected based on the duration of their HIV infection (diagnosis < 5 years vs. diagnosis > 5 years), gender, and type of antiretroviral therapy (dual therapy vs. triple or quadruple therapy). Physicians who followed PLWH at least one day a week and prescribed ART were eligible to participate. The selection of physicians was guided by diversity in professional experience (young vs. older physicians), declared attitude towards dual therapy (favorable vs. reticent), and gender, while the selection of patients was guided by criteria related to their HIV status and treatment. Attention was paid to the representativeness of the profiles. Patients and physicians were recruited by the study investigators through a snowball effect until data saturation [14]. The interviews were anonymized using a unique code consisting of the participant's initials and a number. The study was conducted in both urban and rural settings across France. The present study followed the Consolidated Criteria for Reporting Qualitative Research (COREQ) checklist to ensure rigor and transparency in the reporting of its qualitative research findings.

## Interviews

Two semi-structured interview guides used were based on scientific literature and research questions, which were approved by the scientific committee. The following themes were addressed (Table 1):

The interviews were conducted by three researchers (GR, MD, ASP), either by telephone or face-to-face (at the hospital). Socio-demographic data were collected by an online self-reported questionnaire before the interview. The interviews were recorded and fully transcribed.

Approval has been obtained from the Independent Ethics Committee of Hôpital Foch registered as an IRB (00012437) on February 3, 2021 (n° 21-01-06). Participants were informed and consent was obtained before each interview. The data collected were anonymized through the allocation of an identifier, to ensure data confidentiality.

## Analyses

A lexicometric analysis using Iramuteq software [15] with the following variables for the physicians was used: gender, location, professional seniority, number of patients, specialty, professional status, and declared attitude towards dual therapy. The lexicometric analysis, thanks to class distribution, made it possible to highlight the structures of the speeches and the prevalence of the themes mentioned by the participants. The two analyses, one focused on physicians and the other on patients, were conducted separately and the results were triangulated to cross-reference the data. For the patients, the variables considered were gender, location, HIV duration, current treatment, modality and duration of treatment, the proposal of dual therapy, and their declared attitude towards changing treatment. All interviews were processed together to highlight common structures between the two populations. The specific "lexical worlds" [16] of each group were also identified. The lexicometric analysis breaks down the corpus by context unit (lemmas) based on their occurrences. Top-down hierarchical clustering (TDHC) was used and Chi-2 tests indicated the strength of associations between lemmas and classes [15 – 17]. The TDHC was complemented by similarity analyses to model the classes [17, 18], in particular, on simplification and dual therapy. The distribution of discourse was studied by correspondence factor analysis (CFA). In addition, thematic analysis [19, 20] was used to provide a more comprehensive view of the experiences through a categorical content analysis, aiming to identify the semantic aspect of the discourses [20]. The interviews were imported and coded into the NVivo 12 software. The triangulation [21] of methods ensured the validity of the data and allowed for a more nuanced understanding of the change of treatment to O2DR.

## Results

### Socio-demographic data

Thirty participants were interviewed: 15 physicians and 15 PLWH (S1 and S2 Tables in S1 File). Each participant was assigned an anonymous code consisting of interview number,

**Table 1. Interview guides subject areas.**

| Themes discussed with physicians | Themes discussed with PLWH |
|---|---|
| (1) Relationships with patients<br>(2) Treatment choice and practices at initiation<br>(3) Treatment choice and practices during a change of treatment<br>(4) Treatment reduction practices<br>(5) Perceptions of treatment reduction<br>(6) Perceptions of O2DR<br>(7) Proposals for switching to O2DR | (1) HIV treatment initiation<br>(2) Experiences of treatment change<br>(3) Perceptions of treatment mitigation strategies<br>(4) Perceptions of O2DR<br>(5) Experiences with tapering and oral combination therapy<br>(6) Non-HIV medication practices<br>(7) Relationships with health professionals |

status (clinician or patient), and gender (male or female). Of the physicians, 8 were women. The median age was 55 years [36.2; 57.7]. They had a median of 200 patients in their practice [105; 300]. The number of patients corresponds to their current patient list at the time of the interview. Ten physicians reported favorable attitudes towards O2DR and 5 physicians reported theoretical agreement, but mentioned cautious use. The data includes their gender, location of practice (either in the Île-de-France region or in a province), age, year of thesis, years of HIV practice, number of HIV patients they have treated, speciality, and hospital/university position (hospital practician, university professor-hospital practician). The mean duration of the interviews was 62 minutes, with a standard deviation of 32 minutes.

Of the PLWH, 9 were male. The median time since HIV diagnosis was 20,5 years [30, 2]. Five people were on dual therapy, 10 were on triple therapy, including one person on an intermittent treatment (4/7 days). Eight had been offered switch for dual therapy, 2 of whom were reluctant to accept. Two participants did not meet the clinical criteria for a switch. These patients were included in the sample although they could not benefit from it. This choice was made in keeping with the comprehensive approach in which the study's objectives are rooted. The data includes their anonymization code, gender, location of follow-up (either in Paris or in a province), year of diagnosis, current treatment, and their attitude towards changing their treatment. The mean duration of the interviews was 46 minutes, with a standard deviation of 14 minutes.

## Lexicometric analysis

The lexicometric analysis enabled the corpus to be divided into 7 classes and 91.7% of the corpus was processed. The distribution in classes allows us to see where the dual therapy is located in the discourses. The dendrogram (Fig 1) represents the classes. Additionally, correspondence factor analysis and similarity analysis were conducted (S3-S5 Figs in S1 File). S3 Fig in S1 File shows that the term "dual therapy" is systematically associated with its medical nature (molecule) and the way patients take it. S4 Fig in S1 File. which represents the similarity analysis carried out on the total corpus, illustrates that patients are at the heart of discourses that include treatment-related issues as well as communicational aspects. Tables with examples of speeches referring to each class is available in the supporting information (S6 and S7 Tables in S1 File). The figures in the supporting information have been kept in French.

In the context of understanding the determinants of treatment change, the classes identified through the lexicometric analysis can be named interpreted as follows:

- Class 1 "Treatment and monitoring": This class represents the determinants of overall HIV management, which includes factors such as adherence to treatment, the impact of HIV on daily life, and the importance of regular medical check-ups.

- Class 2 "Social environment and support": This class focuses on the social support available to individuals managing HIV, including support from friends, family, and healthcare professionals.

- Class 3 "Lived experience and difficulties": This class highlights the individual difficulties, encountered both from the perspective of patients, in their experience of infection. But also among clinicians, the difficulties encountered in care.

- Class 4 "Sources of information about treatments": The class highlights the different sources of information mobilized by doctors in particular, about treatments. This class is directly related to the specific information questions proposed during the interview.

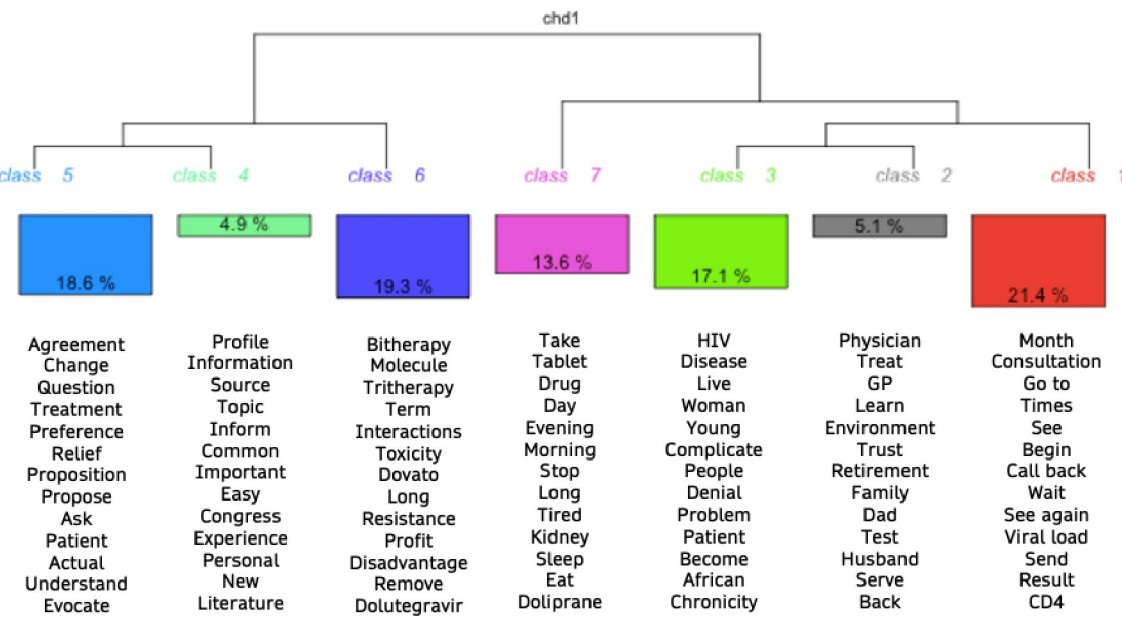

**Fig 1. Dendogram of lexical analysis.**

- Class 5 "Communication about switch": This class represents the process of proposing a treatment change, with a focus on the interaction between physicians and patients. The presence of action verbs and the term "agreement" suggest that patient involvement in the decision-making process is important.

- Class 6 "Scientific vision of the switch": This class also represents medical discourse, specifically the medical vision of dual therapy. This highlights the importance of medical perspectives in the implementation of new treatment options. This class is characterized by medical discourse, specifically the criteria motivating the change of treatment. This suggests that medical factors play a significant role in treatment change decisions.

- Class 7 "Characteristics of HIV chronicity": This class focuses on the characteristics of chronicity, specifically the long-term impact of HIV on individuals' lives and the importance of adapting treatment to these changing circumstances.

Overall, the classes suggest that treatment change decisions are influenced by a range of factors, including medical criteria, social support, and the importance of involving patients in the decision-making process. The lexicometric analysis provides a comprehensive overview of the different themes present in the corpus, highlighting the complexity of the factors involved in treatment change decisions.

### Thematic analysis

The lexicometric analysis was conducted to uncover central patterns in participants' discourse, while the thematic analysis aimed to provide a deeper and more comprehensive understanding of their experiences. Three thematic categories common to both physicians and patients emerged: shared issues between physicians and patients in the context of a change of treatment (1), practices and experiences of O2DR (2), and knowledge of O2DR (3). A table is available in the supporting information (S6 Table in S1 File) describing the topics covered and the associated quotes.

## Shared issues between physicians and patients in the context of a change of treatment

The speeches of physicians and patients highlighted common expectations: maintaining efficacy and tolerance, as well as the integration of the treatment into their daily lives.

"*The idea for me is to have something that has as little impact as possible. I want it to be effective, and in any case, I know I have to live with it. So from this point of view, which it doesn't impact on my daily life, that's the main thing*" (E2-P-M)

Single-tablet regimens (STR) refer to the quality of life and is the most frequently cited criterion by physicians (n = 11) when choosing a treatment. Simplification was also the most significant change mentioned by patients (n = 9). Dual therapy appears here as an opportunity to combine both STR and less drug strategy.

"*So either I want to change treatments to lighten the treatment, or because the treatment is not going well and I will change the treatment. So it's either to lighten the load or to simplify—to make it easier for the patient*" (E7-C-F)

A third of the patients sampled reported that the switch to O2DR was facilitated by proposition from their physician to modernize their treatment (n = 5). The caregivers' discourse (n = 11) also highlighted the desire to improve treatments and to do better for patients as one of their motivations for switching. O2DR when offered for the switch without clinical or biological reasons reflects to some of their professional honesty towards patients.

"*I think it is our duty to re-evaluate the treatments or at least to give information to the patient on the treatments that are being used, on new ones on the market, and inform patients if there are other therapeutic options that could be interesting for them*" (E14-C-F)

In the absence of adverse effects, proposing and being proposed a new treatment is a situation with two issues: fear of intolerance, particularly when the change involves a change in therapeutic classes, and the fear of inefficacy. These shared fears appear to be the main obstacle to changing treatment among both physicians and patients.

The relationship between physicians and patients was commonly mentioned as an important determinant of treatment change. Explanations, pedagogy, and not imposing treatments or adapting to the patient's requests contributed to the acceptance of the proposal. Two elements were highlighted in the discussions: trust in the physician and subjectively, better physician-patient relationship. Trust made it easier for physicians to propose changes in treatment while the patient emphasized that it was because they trusted their physician that they adhered to the change of treatment, especially for those with a long history of HIV.

"*Afterwards, I'm a bit nervous, but if the clinician tells me I should switch to dual therapy, because I'm being monitored anyway, I know there's no risk. I am 100% confident, even more than that!*" (E5-P-F)

## Practices and experiences of O2DR

The decision to accept or propose a change of treatment is based on the individual and therapeutic experiences of physicians and patients. While some of the older physicians indicated that dual therapy was common practice, even before it was marketed as a single-pill regimen,

others preferred to rely on their previous experience with triple therapy before offering O2DR to a patient. For some younger physicians, their professional context encourages them to prescribe dual therapy for their patients.

"*We've been prescribing dual therapy for five years, or maybe even longer, with Isentress®, I'm prescribing dual therapy*" (E10-C-M)

For patients who tolerate their current treatment well, physicians point out a dilemma between theory, wanting to do better, and practice, taking the risk of changing treatment without medical and empirical reasoning. Switching to O2DR is a reasoned practice for most physicians, requiring reflection. They emphasize the importance of giving patients time to reflect, as making the switch is not urgent.

"*And then there is also the thing that drives me, which is that the best is the enemy of the good. That's the thing I say, but sometimes I always worry. A patient is on a triple anti-integrase therapy and today we can't reduce without relying on an anti-integrase. And so, at the end of the day, am I not going to put him in such a situation. . .or is there going to be weight gain? I'm not worried about that, but there is that, not to rush it*" (E12-C-F)

PLWH with a long or complex treatment history were either reluctant or in favor of treatment change. Those reluctant focused on the habit and tolerance of their current treatment. A patient on triple therapy taken four days a week would feel compelled to go back to taking once-a-day regimen. In newly diagnosed patients, dual therapy represents a relief considered at the time of initial prescription and appears to be better accepted. Patients with O2DR are satisfied with it, due to good results and the absence of adverse effects or physical symptoms.

"*When I was first offered dual therapy, it took me about three months to decide to lower the dose. It's not that I didn't want to, but I don't like changing my treatment. It disturbs me and I don't understand the benefits, etc., well I don't accept it like that*" (E15-P-F)

"*I was quite satisfied because it was one active ingredient less. But I didn't have any fear of non-efficiency. She suggested it because, in light of my tests, she thought we could switch to dual therapy, and I was not at all opposed to it*" (E14-P-M)

The change in treatment towards a less drugs strategy among PLWH is also to be considered in the context of the global relationship with health and medication practices. Most of the patients declared a reasoned practice of taking medication apart from HIV treatment. But for a minority, the experience of HIV contributed to over-medication, reflecting the fear of becoming ill. Three of the participants emphasized the use of alternative medicines such as naturopathy or homeopathy. Overall, their relationship with medication corresponded to the desire to preserve their health and to be able to conduct their daily activities. In this context, for some people, less drug strategy makes sense to protect their health and their body in the long term.

"*[. . .] at the beginning I was taking shit. It made me feel worse inside, I guess, a bit more than if I had taken a dual therapy from the start*" (E2-P-M)

## Knowledge of O2DR

For physicians, O2DR has no specific characteristics that would allow it to be used systematically. Although all physicians knew treatment guidelines, only two of them used dual therapy

as a first prescription. It seemed more reassuring for physicians to start with triple therapy, and to keep dual therapy as one of solutions for therapeutic simplification.

"*The feeling I have is that [dual therapy] is really an additional weapon to adapt the treatment as well as possible, so that there are fewer side effects and less toxicities*" (E2-C-M)

The physicians (n = 5) identified as being more cautious with O2DR highlighted certain limitations of dual therapy: a lack of empirical evidence on the prevention of toxicity, reservoirs of resistance and potential risks of failure, and limited choice of molecules marketed in the form of a single tablet. These factors may hinder them from prescribing O2DR, preventing them from fully reassuring their patients.

"*There is a financial benefit, which is not hypothetical. Apart from the reduction in long-term toxicity for the patient, it remains hypothetical*" (E8-C-M).

On the patient side, most had heard about O2DR from their physician. Three patients needed explanation on O2DR therapy signification. The patients reported a common view of treatment reduction, perceived as an evolution in the history of HIV treatment, particularly among patients who have been infected for more than five years (n = 10).

"*Yes, because each time it's a new molecule, so it's progress! The laboratories are not going to put on the market drugs that are less effective or less easy to take!*" (E6-P-M)

The patients' views were in line with those of the physicians. Preventing toxicity is translated in patients' discourse as preserving health and reducing the aggression on their body. O2DR was also synonymous with good health: the possibility of lightening the treatment means they are healthy. Three patients, with long HIV duration highlighted a fear of loss of efficacy, outweighing the benefit of preventing kidney toxicity.

"*If it's less aggressive on the organs, and I can take it because I'm better, everything is fine*" (E13-P-M).

Various sources of information contributed to representations about dual therapy. Physicians emphasized multiple sources, such as congresses, scientific literature, clinical staff, and exchanges between peers. Physicians are the main source of information for most patients. Few of them (n = 4) sought information on new treatments, mostly via specialized websites.

Finally, the representations of O2DR highlight comparisons with other relief strategies, notably 4/7-days intermittent treatment and injectable treatments which were not yet on the market at the time of the study. Among physicians, when compared with an intermittent treatment, O2DR was perceived as a safer strategy because of its daily dosing regimen. Compared to dual therapy, intermittent treatment seems to be proposed less spontaneously by physicians. While patients did not request O2DR, they seemed to ask for intermittent treatment, especially younger patients, some even tried it without prior discussion with their physician In the discourse of both patients and clinicians, O2DR is compared to injectable treatment, which is also a dual therapy. On this point, the results underline the fact that injectable treatments are perceived as more innovative than oral dual therapy.

"*I easily offer dual therapy. I sometimes do 4/7 days a week, I have a few but it's more likely to be at the patient's request.*" (E15-C-F)

Two of the patients with intermittent treatment were satisfied, as it offers a weekly break which seemed to improve their psychological quality of life. For the others, the mention of reductions of doses highlights two types of attitudes. Those are reluctant to reducing their weekly dose, stressed a fear of forgetting the treatment and seeing their viral load increase. Once-a-day regimen was not perceived as a constraint for these patients. While those who discovered this regimen during the interview, they expressed interest in it. Moreover, as with O2DR, intermittent strategy was perceived as an advance in medicine.

"*I think I'm fine with my treatment, and it gives me a holiday two days a week. Sometimes I go to parties with friends, and I don't really think about taking it with me, things like that.*" (E10-P-F)

O2DR is also compared with injectable treatments. Just like intermittent treatment, the physicians mentioned a request from the patients, contrary to O2DR, and most of them (n = 9) were motivated to propose an injectable treatment. The speeches highlighted the innovative perspective in terms of administration, injectable treatments bring an added value for the quality of life of patients as well as an interesting alternative for non-compliant patients. Some physicians anticipated patients' constraint of going to the hospital.

"*Injectable dual therapy is a plus because some patients are interested in having more freedom, more anonymity or in any case, more confidentiality in relation to their environment to have an injectable treatment*" (E14-C-F)

## Results, discussion, conclusions

### Main results

Our results show that O2DR is proposed, more or less close to the HIV infection diagnosis and, depending on the patient's history. Even though it is recommended for first line prescription, some practitioners seem to be reluctant mainly due to ingrained prescribing habits. A long medical practice in HIV is determinant for changing to dual therapy. Physicians with longer experience appear to be more reluctant to prescribe dual therapy as a first prescription and generally show more hesitancy before proposing a switch. Among PLWH, those with longer HIV history seem to be more apprehensive about efficacy. The presence of common motivations contributes to the acceptability to switch treatment for a dual therapy: simplification, a single tablet, maintaining efficacy and good tolerability, the desire to modernize and to have "better". The proposal and acceptability of a change of treatment to dual therapy stems from a process of comparison with other strategies for reducing treatment. The representations and knowledge of O2DR contribute to the acceptability of the treatment change. Physicians approach O2DR from a medical and scientific characteristics point of view, allowing a treatment change to be proposed. Conversely, patients do not elaborate ideas about O2DR, it is thought of as a medical advance, and when it has been proposed to them, it is associated with a good health. The acceptability of switching to dual therapy depends on the care relationship, where it appears that trust is central to the proposal. Physicians emphasize that patients' trust make it easier to propose. Patients rarely seek information and trust their physician's proposal, the trust reflecting an external control of patient.

### Triangulation and comparisons with the literature

The triangulation of the methods [21] highlights three cross-cutting axis in the acceptability of the change of treatment.

**Evaluation process of O2DR.** The acceptability of switching to dual therapy is the result of a decision-making process. Comparison and evaluation are cognitive determinants of the decision-making process [22], as are the level and sources of information. Among physicians, the proposal and acceptability of change is the result of a rational decision-making process [23], guided by efficacy. Physicians focus on the clinical and biological criteria, patients' living conditions, and preferences. They rely on scientific information but also on their personal beliefs, along with previous prescribing experiences. Studies [23–25] highlight that the nature of the reasons for switching treatment influence the decision process. Our results highlight the reasons associated with the treatment change are variable. In the case of side effects, patients appear to be pro-active in requesting a treatment change to dual therapy or for simplification [26]. Two studies [27, 28] showed similarly that side effects and simplification are the main motivations for switching to O2DR. In our study, the motivations are aimed at a theoretical desire to do better, despite the absence of symptoms declared by the patients. The process of comparing O2DR with other therapeutic relief strategies participates in the reflexive process of changing treatment, through a benefit-risk balance. For physicians, O2DR is perceived as more reassuring than intermittent treatment, and in practice, O2DR is more often prescribed by physicians than intermittent treatment. For patients, the intermittent treatment in the number of doses compared to O2DR highlights the fear of forgetting. On the other hand, O2DR appears to be a constraint. When O2DR is compared to long-acting injectable treatments, the more innovative way of administration is widely emphasized.

**Physician-patient relationships: The challenge of trust.** Although the objective and cognitive factors are claimed, importance of the care relationship for treatment change is crucial. The lexicometric and thematic analyses have highlighted elements that were a priori contradictory. Firstly, class five shows the importance of the patient's agreement and preferences in the context of treatment change, suggesting an active process of shared medical decision making [29]. Nevertheless, thematic analysis of physician and patient discourse shows that trust tends to make the physician the main decision maker [30]. Trust, in the context of healthcare relationships, is a determining factor in patients' ability to make decisions about their health and in their compliance to treatment. Trust eliminates uncertainty, implies stability, and suggests the presence of resources to effectively deal with any risk [31]. In the context of our study, trust should be considered from two additional perspectives: recognition of knowledge and management of control. Most patients are not searching for information, and perceive the physician as the bearer of knowledge they do not have, revealing a dichotomy between expert knowledge and lay knowledge [32]. In our results, the physician appears to be the bearer of expert knowledge, being the main or only source of information. These findings are consistent with the results of another study [33] which showed that the trust created in the physician-patient relationship depends on the quality of care, communication, and the perceived physician's competence [34]. Our results show that with the feeling of trust, most patients tend to put treatment choice in the hands of the specialist and to adhere to their proposal. Physicians reported the proposal easy thanks to patients' confidence. The notion of confidence present in our results highlights that the proposal of a treatment change to dual therapy seems to depend on an external perception of control. The psychological locus of control (LOC) model conceptualizes the two ways in which individuals view their actions on life events and refers to their beliefs about their personal control over their environment [35]. The internal LOC refers to beliefs about self-determination, and that it is their behaviors, abilities and attributes that enable them to cope with their environment. The external LOC refers to individuals who believe that they have little control, and that decisions depend on more powerful people then themselves. According to this definition, it appears that trust, as a shared finding between physicians and patients in our study, is materialized by a transfer of decision from the patient to

the physician. Confidence is a driving force for adherence to treatment change and patients consider the physician as an effective control figure for the disease [36, 37]. Consequently, patients mostly accept physicians' proposal. In contrast, in our study, the practices are circumscribed to the framework of the care relationship, where compliance with medical choices and decisions through a dynamic of knowledge recognition is central, particularly in changing treatment.

### Limitations and strengths of the study

One strength of our study is its originality as no study has been published on the determinants of acceptability of a treatment change in the context of O2DR. The triangulation of methods allowed us to highlight that the acceptability of treatment change to O2DR is multifactorial and sheds light on issues relating to the care relationship. We have identified some limitations. Firstly, even though we recruited a diverse range of patient profiles, no patients with long experience of O2DR were included in the study to provide more information from their experience. In addition, some patients could be more aware of dual therapy as they participated in the study. Although we encountered physicians who were reluctant to use dual therapy, interviewing practitioners who did not prescribe it could have identified other determinants.

### Conclusion

The study shows that the acceptability of a change of treatment to O2DR depends on motivations related to the decreased risk of adverse effects and simplification. It also depends on the professional experience of the physicians and the HIV duration. Proposing a treatment change for physicians and accepting it for patients is a reasoned process of evaluation and comparison, motivated specifically by the desire to "do better" and "have better". Finally, the framework of the care relationship, and in particular, trust, appear to be important levers.

### Perspectives

The PROBI qualitative phase has enabled the creation of flyers for physicians and patients to guide and inform them about the change of treatment to O2DR.

Finally, the results, particularly concerning the comparison of therapeutic relief strategies, open perspectives for studies on the prioritization of lower drug consumption regimes.

### Supporting information

**S1 File.**
(DOCX)

### Acknowledgments

We would like to thank all research participants, including all the centers that participated in the PROBI study: Marseille European hospital, Foch Hospital, Nantes University Hospital, Necker Hospital, Hotel-Dieu Hospital, Bicêtre Hospital, Reims University Hospital, Saint-Denis Hospital, La Roche sur Yon Departmental Hospital, GHT Atlantique, Poitiers University Hospital, Guadeloupe University Hospital, Orléans Regional Hospital Centre, Clermont-Ferrand University Hospital, Rennes University Hospital, Sainte Marguerite Hospital, Rouen University Hospital, Jean Verdier Hospital, Sud Francilien Hospital, Villeneuve Hospital, Toulon Hospital, Créteil Hospital, Antoine Béclère Hospital and Caen Universiry Hospital. We would like to thank all technicians in clinic studies: Nouara Agher, Morane Cavellec, Françoise Churaqui, Caroline Debreux, Barbara De-Dieuleveult, Jean-Charles Duthe, Emelyne Duvallon,

Barbara Gasse, Emilie Goncalves, Isabelle Kmiec, Laetitia Laine, Véronique Lambry, Manuela Le Cam, Randa Maamar, Lucie Marchandeau, Flory Mfutilakaykay, Ernesto Paredes-Manyari, Marie-Pierre Pietri, David Plainchamp, Sandrine Poirier, Anne Ricci, Laurent Richier, Ketty Samar, Albane Soria, David Theron, Fatima Touam, Guillemette Unal and Camille Vassord-Dang. We are grateful for URC-Eco team's feedback. We thank Jose M Valderas for his feedback on the results of the qualitative study.

**Scientific committee**

Clotilde Allavena, MD, Nantes

Olivier Chassany, PU-PH, Paris

Martin Duracinsky, MD, Paris

Laurent Hocqueloux, MD, Orléans

David Zucman, MD, Suresnes

## Author Contributions

**Formal analysis:** Anne-Sophie Petit.

**Investigation:** Martin Duracinsky.

**Methodology:** Anne-Sophie Petit, Guillaume Roucoux, Martin Duracinsky.

**Project administration:** Clotilde Allavena, David Zucman, Laurent Hocqueloux, Olivia Rousset-Torrente, Olivier Chassany, Martin Duracinsky.

**Resources:** Clotilde Allavena, David Zucman, Laurent Hocqueloux, Olivia Rousset-Torrente, Guillaume Roucoux, Claudine Duvivier, Gwenaël Le Moal, Martin Duracinsky.

**Supervision:** Clotilde Allavena, David Zucman, Laurent Hocqueloux, Olivia Rousset-Torrente, Olivier Chassany, Martin Duracinsky.

**Validation:** Clotilde Allavena, David Zucman, Laurent Hocqueloux, Olivia Rousset-Torrente, Claudine Duvivier, Gwenaël Le Moal, Olivier Chassany, Martin Duracinsky.

**Writing – original draft:** Anne-Sophie Petit, Clotilde Allavena, David Zucman, Laurent Hocqueloux, Olivia Rousset-Torrente, Guillaume Roucoux, Claudine Duvivier, Olivier Chassany, Martin Duracinsky.

**Writing – review & editing:** Anne-Sophie Petit.

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
