## [Decision Letter · Decision Letter 0]

12 Dec 2022

PONE-D-22-15158Acceptability determinants of a proposal to reduce antiretroviral treatment to an oral two-drug regimen among patients and physicians in France.PLOS ONE

Dear Dr. Petit,

Thank you for submitting your manuscript to PLOS ONE. After careful consideration, we feel that it has merit but does not fully meet PLOS ONE’s publication criteria as it currently stands. Therefore, we invite you to submit a revised version of the manuscript that addresses the points raised during the review process.

 Your manuscript has been assessed by two peer-reviewers and their reports are appended below.  The reviewers comment that aspects of this study should be strengthened by additional detail or further clarification, for example the methodology and the results sections. Furthermore, one of the reviewers comments that the lexicometric analysis presented in this study could be developed further, and that quotations should be presented to support the study's findings.  Could you please revise the manuscript to carefully address the concerns raised?

We look forward to receiving your revised manuscript.

Kind regards,

Maria Elisabeth Johanna Zalm, Ph.D

Editorial Office

PLOS ONE

Journal Requirements:

3. Thank you for stating the following in the Competing Interests:

“I have read the journal's Policy and the authors of this manuscript have the following competing interests: CD has received consultancies, speaker honoraria and travel grants from Gilead, Janssen-Cilag, MSD and Viiv Healthcare ; GM has received fees for participation in advisory boards or presentations from Gilead and MSD ; CA has received travel grants and honoraria from Gilead, Viiv Healthcare and MSD ; LH has received fees for participation in advisory boards or présentations from Viiv Healthcare and Merck, LH has received travel support for congress attendance from Viiv Healthcare, Merck, Gilead and Shape and Dohme ; MD has received consultancies, speaker honoraria and travel grants from Gilead, MSD and Viiv Healthcare ; OC has received consultancies and speaker honoraria from Gilead and Viiv Healthcare.

The following authors have declared that no competing interests exist : ASP, OT, GR and DZ.”

We note that one or more of the authors have an affiliation to the commercial funders of this research study : Viiv Healthcare, Merck, Gilead and Shape and Dohme

4. Please include a caption for figure 1.

Additional Editor Comments (if provided):

Please carefully review the person-identifiable information provided in the tables in this study, and consider whether some of the individual level data presented in these figures could be further anonymised into subgroup brackets (for example age [30-34], [35-39], years of practise [1-5], [6-10], [11-20] etc). Please keep in mind that study participants may be identified with as little as 3 pieces of person identifiable information (such as age, year graduated/diagnosed, location of practise, etc).    

Reviewers' comments:

Reviewer's Responses to Questions

**Comments to the Author**

1. Is the manuscript technically sound, and do the data support the conclusions?

Reviewer #1: Yes

Reviewer #2: Partly

2. Has the statistical analysis been performed appropriately and rigorously? 

Reviewer #1: I Don't Know

Reviewer #2: N/A

3. Have the authors made all data underlying the findings in their manuscript fully available?

Reviewer #1: Yes

Reviewer #2: Yes

4. Is the manuscript presented in an intelligible fashion and written in standard English?

Reviewer #1: Yes

Reviewer #2: Yes

5. Review Comments to the Author

Reviewer #1: line 52-54: Please elaborate the connection of this study to the PROBI qualitative study.

line 61-63: I think this should be described more straightforward to avoid confusion. In addition, the location of practice should be described in a simpler fashion, perhaps urban vs. rural, capital vs. provincial, etc, so that it's easier for readers not-so-familiar with France.

Table 2: please consider presenting the summary instead of individual data and see which works best. Also, please provide explanation on abbreviations.

Table 3: would it be possible to include the patients' sociodemographic characteristics as well, e.g. education, employment status, etc. Also, consider presenting them in summary, not individual data.

Figure 1: this can be improved by adding, for instance: describe what class 1-7 refers to, put frequency (%) to the words, etc.

Reviewer #2: First I would like to congratulate the authors on this manuscript. Thank you for the opportunity to review this work. This manuscript presents the results of a qualitative study investigating the determinants of the acceptance of a proposal to reduce antiretroviral treatment in patients living with HIV and physicians. This work brings to light very interesting results which could help better care for patients and relies on a double analysis: statistical text analysis and thematic analysis. However, the introduction is quite short and the statistical analysis appears quite secondary and should be better described/elaborated on. Similarly, I think the methods and interpretation part of this study should be better described to be fully understood and assessable.

Major comments

Introduction

1. "to date there lacks studies aimed at questioning the psychosocial determinants at stake in the context of treatment change" : Do the authors mean there is not any study on this topic? If so it should be worded that way.

2. The introduction is not enough supported. In my opinion, the authors should develop it a bit more. I may suggest to discuss the literature on treatment adherence and/or shared decision making and/or other connected topics for example?

Methods

1. The authors specify they proceeded to a theoretical sampling logic. Were there any theoretical recruitment process (i.e., ideal proportion of participants according to each identified characteritic)?

2. Similarly, their sampling was based on criteria thought of as indicators of the phenomena. How did the authors decided that these variables were indicators? This should be broached upon in the introduction.

3. How was the professional experience dichotomized? What was the threshold? Why?

4. The authors mention recruitment continued till data saturation. How was it assessed?

5. The interview guides were quite on the "directive side" of semi-structured interviewing, why did the authors choose to analyze conjointly the patients' and physicians' interviews? I am quite surprised for this choice as interview guide may both influence the topics told by participants and the word they use to tell their experience. Separate analyses may be a more sound choice.

6. P.3 The authors specify the variables taken into account in the Iramuteq analysis. What were the modalities of the "sexual orientation" variable for example? The results section do not present results pertaining to these variables. I advise the authors to either not present these variables in the method section or present the results associated with them.

7. The lexicometric analyses (as well as the interpretation process) should be more thoroughly described as it is not a well-known kind of analysis.

8. The authors specify they proceeded to a thematic analysis/categorical content analysis. Was it an inductive or deductive one? Were there any double-coding involved? If so, I advise the author to provide an intercoder coefficient.

9. More generally, I advise the authors to use a quality checklist (e.g., COREQ) to ensure rigour in their study.

Results

1. The authors specify the number of patients in physicians' practice. Does this concern the current or passed pratice?

2. It is unclear to me whether the two participants who did not meet the clinical criteria for a switch were included in the study or not.

3. How were patients classified as "adherent", "hesitant" or "opposed"? What was the authors' criteria?

4. What was the mean (+ standard deviation) of the interviews' time?

5. Offering a translation for the appendices to make them fully accessible to all readers seem relevant to me.

6. Regarding the lexicometric analysis:

a. The authors mention that classes 4, 5 and 6 are "saturated" with medical discourse. Do they mean it as a kind of "data saturation"?

b. To me this part is not enough developed. Each class should be named, their interpretation should be argued and quotations should be presented to ensure credibility (Lincoln & Guba, 1985)

7. P.5 L.150 : "three thematic categories common to both physicians an patients", do the authors mean that both groups were in agreement on these aspects?

8. Ideally (and if the word limit allows it), when an idea refering to opinions found in both patients and professionals is presented, quotations from both of these groups should be presented.

9. As the results pertaining to the lexicometric analysis are way less developped compared to the thematic analysis I wonder what was the added value of this kind of analysis according to the authors?

10. I found the discussion of the article very interesting to read but the authors added some results that were not presented in the results section (e.g., P.10, L.384-386), these information should be first be well presented in this part.

Minor comments

1. p.2 l.67 : ARV is written as an acronym while it is the first time the authors mention it. They should write it fully here.

2. p.2 l.80 "themes" to describe "topics" broached upon in the interview guide is not the best term as it get mixed up with "themes" resulting from a thematic analysis.

3. Quotations for both the lexicometric and thematic analyses could be presented in a table to respect the word count without compromising on the content of their manuscript.

4. The age, gender and status (patients vs. professional) should be specified for each quotation.

5. "Single-tablet regimens (STR) refer to the quality of life" do the authors mean "refer" or "guarantee" (or something else)?

6. There are some typos in the manuscript (e.g., P.6 L. 191 "a subjectively, a better")

7. The title "Results, discussion, conclusions" seems inappropriate.

6. PLOS authors have the option to publish the peer review history of their article (what does this mean?). If published, this will include your full peer review and any attached files.

Reviewer #1: No

Reviewer #2: **Yes: **Lucile Montalescot

---

## [Author Response · Author response to Decision Letter 0]

3 Nov 2023

Thanks to the reviewers for the changes made to the article. A response document to each question was provided upon resubmission.

---

## [Decision Letter · Decision Letter 1]

17 Jan 2024

PONE-D-22-15158R1Acceptability determinants of a proposal to reduce antiretroviral treatment to an oral two-drug regimen among patients and physicians in France.PLOS ONE

Dear Dr. Petit,

Thank you for submitting your manuscript to PLOS ONE. After careful consideration, we feel that it has merit but does not fully meet PLOS ONE’s publication criteria as it currently stands. Therefore, we invite you to submit a revised version of the manuscript that addresses the points raised during the review process.

We look forward to receiving your revised manuscript.

Kind regards,

Ivan Alejandro Pulido Tarquino, MSc

Academic Editor

PLOS ONE

Journal Requirements:

Additional Editor Comments:

Dear Authors,

Please, ensure that all revisions are consistent with the rest of the manuscript. Each section should be clear, concise, and convey the intended message without ambiguity.

Verify that all statistical data, including median and IQR values, are accurate and clearly presented. Methodological details, especially regarding analysis techniques like lexicometric analysis, should be explicitly explained for the benefit of readers who may not be familiar with these methods. Ensure that all terms, acronyms, and references are consistently used and clearly defined at their first occurrence. This is especially important for acronyms like PROBI, which might not be universally recognized.

Reviewers' comments:

Reviewer's Responses to Questions

**Comments to the Author**

1. If the authors have adequately addressed your comments raised in a previous round of review and you feel that this manuscript is now acceptable for publication, you may indicate that here to bypass the “Comments to the Author” section, enter your conflict of interest statement in the “Confidential to Editor” section, and submit your "Accept" recommendation.

Reviewer #1: All comments have been addressed

Reviewer #3: (No Response)

2. Is the manuscript technically sound, and do the data support the conclusions?

Reviewer #1: Yes

Reviewer #3: Yes

3. Has the statistical analysis been performed appropriately and rigorously? 

Reviewer #1: Yes

Reviewer #3: Yes

4. Have the authors made all data underlying the findings in their manuscript fully available?

Reviewer #1: Yes

Reviewer #3: Yes

5. Is the manuscript presented in an intelligible fashion and written in standard English?

Reviewer #1: Yes

Reviewer #3: Yes

6. Review Comments to the Author

Reviewer #1: (No Response)

Reviewer #3: Dear Authors, it is a privilege to be part of the review process and contribute to the enhancement of such a promising study.

I find the study particularly interesting, especially considering the chosen topic, which has been relatively underexplored thus far. The application of a mixed-method approach adds a valuable dimension to the research, aligning with the evolving trends in the field.

Having reviewed the manuscript, I appreciate the effort that has gone into addressing the comments raised by other reviewers.

I have minor revisions as request in order to further improve the quality of the manuscript.

Minor revisions:

Title: For clarity, I would suggest adding HIV in the title.

Abstract:

Line 9: I would suggest again to indicate “treatment for HIV”.

Line 12: I would suggest a colon instead of full stop “The study includes 30 interviews: 15….”

Line 14: I would suggest merging and rephrase the following two sentences to give a clearer information about the double analysis. “A double analysis was carried out on the whole corpus. Lexicometric and thematic analyses made it possible to highlight the structuring of discourses 16 around treatment change and the associated issues in the context of O2DR.”

Introduction:

Line 64-66: I would specify that the “two oral O2DR have been approved by health authorities” from Europe and North American.

Line 52-59 : Could you specify what PROBI means? I guess it is an acronym.

Line 81: To be consistent, I would suggest to not use the word PROBI qualitative “study” since you are presenting the PROBI study as a mixed methods study. Please, keep referring to each quantitative and qualitative part as a component or a phase rather than as a study.

Methods:

It would be interesting to have more information about the kind of health facilities where both physicians and patients were recruited since in other countries the “prise en charge” of HIV patients can be realised in different services/settings.

Line 124: I would avoid terms like “the ideal proportion”.

line 130: Please add ref for the COREQ checklist.

Line 153: To make the methodology clearer, I would suggest adding a brief sentence explaining the purpose of the lexicometric analysis.

Results:

Line 181-2 Please, rephrase the sentence “The number of patients in the physicians' practice is their current practice and active practice at the time of the interview” is not clear.

Line 189: The median time since HIV diagnosis was 21 years [30, 1], the IQR is uncorrect

Line 193-194: the reason of the “exception” is not clear.

7. PLOS authors have the option to publish the peer review history of their article (what does this mean?). If published, this will include your full peer review and any attached files.

Reviewer #1: No

Reviewer #3: No

---

## [Author Response · Author response to Decision Letter 1]

27 Feb 2024

Review Comments to the Author

Reviewer #1: (No Response)

Reviewer #3: Dear Authors, it is a privilege to be part of the review process and contribute to the enhancement of such a promising study.

I find the study particularly interesting, especially considering the chosen topic, which has been relatively underexplored thus far. The application of a mixed-method approach adds a valuable dimension to the research, aligning with the evolving trends in the field.

Having reviewed the manuscript, I appreciate the effort that has gone into addressing the comments raised by other reviewers.

I have minor revisions as request in order to further improve the quality of the manuscript.

Minor revisions:

Title: 

For clarity, I would suggest adding HIV in the title.

Thank you for your feedback. The title has been changed as follows: Acceptability determinants of a proposal to reduce antiretroviral treatment to an oral two-drug regimen among patients living with HIV and physicians in France.

Abstract: 

Line 9: I would suggest again to indicate “treatment for HIV”.

We have changed the sentence as follows on line 9 : An oral two-drug regimen (O2DR) in the form of a once-a-day single tablet is now recommended for treatment switching and treatment initiation for HIV.

Line 12: I would suggest a colon instead of full stop “The study includes 30 interviews: 15….”

The change was made to line 13 as follows: The study includes 30 interviews : 15 were conducted with doctors caring for people living with HIV, 15 were conducted with patients who had been offered a change of treatment.

Line 14: I would suggest merging and rephrase the following two sentences to give a clearer information about the double analysis. “A double analysis was carried out on the whole corpus. Lexicometric and thematic analyses made it possible to highlight the structuring of discourses 16 around treatment change and the associated issues in the context of O2DR.”

We have taken your comment into account. The sentence has been changed on line 14 for greater clarity: A double analysis was carried out: lexicometric analysis to highlight the structuring of the discourse around the change in treatment and a thematic analysis to understand the associated issues more precisely.

Introduction:

Line 64-66: I would specify that the “two oral O2DR have been approved by health authorities” from Europe and North American.

Nous avons rajouté cette précision à la ligne 59 de la manière suivante : Also, the switch to O2DR has been approved by health authorities from Europe and North American.

Line 52-59 : Could you specify what PROBI means? I guess it is an acronym

It is in fact an acronym. For greater clarity, we have defined the acronym PROBI in the introduction on line 12.

Line 81: To be consistent, I would suggest to not use the word PROBI qualitative “study” since you are presenting the PROBI study as a mixed methods study. Please, keep referring to each quantitative and qualitative part as a component or a phase rather than as a study.

Thank you for your feedback. For greater consistency, we have taken your suggestion into account by using the term phase rather than study in the corresponding lines (61 and 66) and this has been standardised in the article.

Methods:

It would be interesting to have more information about the kind of health facilities where both physicians and patients were recruited since in other countries the “prise en charge” of HIV patients can be realised in different services/settings.

Thank you for your comment, these are hospital centers. We have specified the centers from which the participants in lines 75 and 77 were recruited.

Line 124: I would avoid terms like “the ideal proportion”.

We have changed the term. The modified sentence is as follows line 89 : Attention was paid to the representativeness of the profiles. 

Line 130: Please add ref for the COREQ checklist.

We have made a reference in the text to the corresponding appendix line 144. 

Line 153: To make the methodology clearer, I would suggest adding a brief sentence explaining the purpose of the lexicometric analysis.

Following your feedback, we have proposed an explanatory sentence to introduce the methodology. The modified sentence is on line 118: The lexicometric analysis, thanks to class distribution, made it possible to highlight the structures of the speeches and the prevalence of the themes mentioned by the participants.

Results:

Line 181-2 Please, rephrase the sentence “The number of patients in the physicians' practice is their current practice and active practice at the time of the interview” is not clear.

Thank you for that clarification. We preferred the term active patient list for greater clarity. The sentence has been changed on line 146: The number of patients corresponds to their current patient list at the time of the interview. 

Line 189: The median time since HIV diagnosis was 21 years [30, 1], the IQR is uncorrect

Thank you for your feedback, there was indeed an error in the calculation of the median. The calculation has been corrected on line 154

Line 193-194: the reason of the “exception” is not clear.

We're sorry, but we can't see what exception is being referred to.

---

## [Decision Letter · Decision Letter 2]

17 May 2024

PONE-D-22-15158R2Acceptability determinants of a proposal to reduce antiretroviral treatment to an oral two-drug regimen among patients living with HIV and physicians in France.PLOS ONE

Dear Dr. Petit, Thank you for submitting your manuscript to PLOS ONE. After careful consideration, we feel that it has merit but does not fully meet PLOS ONE’s publication criteria as it currently stands. Therefore, we invite you to submit a revised version of the manuscript that addresses the points raised during the review process.

**ACADEMIC EDITOR: ** Firstly, on behalf of PLOS ONE I would like to apologise for the extensive time this review has taken so far. It was a bit difficult to find an available reviewer with the expertise required by your piece of work.I am delighted to share with you the comments of the reviewer. In providing your response, I strongly suggest taking into consideration the suggestion to include the perspectives of both participants and physicians, as the reviewer highlighted. Indeed, this would add a wider and more detailed picture of the themes addressed in the manuscript. As you may know if it is important to understand how beneficiaries understand health policies, strategies, and care procedures, on the other end it is also relevant to have an idea about how healthworkers perceive the needs and questions of their patients.  Please address any further comments properly.Thank you

We look forward to receiving your revised manuscript.

Kind regards,

Ivan Alejandro Pulido Tarquino, MSc

Academic Editor

PLOS ONE

Journal Requirements:

Reviewers' comments:

Reviewer's Responses to Questions

**Comments to the Author**

1. If the authors have adequately addressed your comments raised in a previous round of review and you feel that this manuscript is now acceptable for publication, you may indicate that here to bypass the “Comments to the Author” section, enter your conflict of interest statement in the “Confidential to Editor” section, and submit your "Accept" recommendation.

Reviewer #3: All comments have been addressed

Reviewer #4: All comments have been addressed

2. Is the manuscript technically sound, and do the data support the conclusions?

Reviewer #3: Yes

Reviewer #4: Yes

3. Has the statistical analysis been performed appropriately and rigorously? 

Reviewer #3: Yes

Reviewer #4: Yes

4. Have the authors made all data underlying the findings in their manuscript fully available?

Reviewer #3: Yes

Reviewer #4: No

5. Is the manuscript presented in an intelligible fashion and written in standard English?

Reviewer #3: Yes

Reviewer #4: Yes

6. Review Comments to the Author

Reviewer #3: Dear authors,

I wanted to extend my appreciation for your response to the comments I provided on your manuscript.

After reviewing your revisions, I am pleased to confirm that you have addressed all of the concerns and suggestions I raised.

I commend your efforts and believe that the revisions have significantly strengthened the manuscript.

Best regards

Reviewer #4: I appreciate the opportunity to review this manuscript and would like to reiterate previous reviewers’ comments that the manuscript is interesting, well-written, and methodologically sound. I am somewhat surprised that the authors chose to submit this manuscript to PLOS One rather than a specialized journal focused on HIV and/or qualitative research. Nevertheless, articles in PLOS One undergo rigorous peer review and tend to receive wider exposure, which, from my personal experience, can be beneficial. In my opinion, the authors have carefully and thoroughly addressed previous reviewers’ comments; therefore, I do not believe acceptance should be delayed further. However, I have some minor suggestions that could potentially improve the manuscript. Whether or not the authors decide to implement these changes should not affect the manuscript’s suitability for publication.

Tout simplement, je cherche des poux.

Supplementary information

1. The tables in the supplementary information file are not presented in sequential order. Notably, there are two instances labelled as “Table 3”, which could lead to confusion. I would suggest that the authors carefully review their table captions and make the necessary amendments to their manuscript, ensuring that tables are properly referenced.

2. The acronyms (e.g., HP, UP-HP) and terms (e.g., adhere, hesitant, opposed) used in the tables are not defined in the caption. Tables should be self-explanatory and comprehensible on their own. I recommend that the authors include definitions for these acronyms and terms within the table captions (or footnotes) to improve readability and understanding without necessitating reference to the main text.

3. I recognize the challenges associated with translating the similarity analysis (bi-therapy and total corpus) from French, particularly in preserving the original meanings conveyed by the interviewees. However, the figure captions currently lack comprehensive descriptions, making them difficult to interpret independently. I suggest that the authors provide more detailed descriptions of the figures, including the principal words/themes identified. This would make the figures more accessible and informative, especially for non-French speaking readers, allowing them to benefit from the insights conveyed.

4. The sociodemographic data presented, particularly concerning the patients, is quite limited. I would like to know what type of information was collected through the online self-reported questionnaire prior to the interviews. Comprehensive sociodemographic data are essential for contextualizing the findings, as the acceptability of switching to once-daily regimens (O2DR) may differ significantly based on individual lived experiences. For instance, racial and ethnic disparities exist along the HIV care continuum, and the perspectives of people of color living with HIV may vary from those of other key populations (ex: white MSM, PWID). This could also have programmatic implications. If the authors have chosen not to present this data, it is crucial that this decision be explicitly mentioned in the manuscript as a limitation of the study.

Main text

1. The authors have chosen not to include opinions common to both patients and physicians, citing word limit constraints as the reason. However, it’s important to note that PLOS One does not impose a word count limit. I encourage the authors – and the Editor – to use the space needed to fully present themes and ideas, particularly those supported by both patient and physician perspectives. Qualitative and mixed-method studies often require more extensive discussion to thoroughly explore concepts and results. I highly recommend that the authors reconsider their approach if possible.

7. PLOS authors have the option to publish the peer review history of their article (what does this mean?). If published, this will include your full peer review and any attached files.

Reviewer #3: **Yes: **Valentina Carnimeo

Reviewer #4: No

---

## [Author Response · Author response to Decision Letter 2]

1 Jul 2024

Dear Reviewer,

Thank you for your thoughtful and positive feedback on our manuscript. We appreciate your recognition of the manuscript's quality and your acknowledgment of our efforts in addressing previous reviewers' comments.

---

## [Decision Letter · Decision Letter 3]

31 Jul 2024

Acceptability determinants of a proposal to reduce antiretroviral treatment to an oral two-drug regimen among patients living with HIV and physicians in France.

PONE-D-22-15158R3

Dear Dr. Anne-Sophie Petit

We’re pleased to inform you that your manuscript has been judged scientifically suitable for publication and will be formally accepted for publication once it meets all outstanding technical requirements.

Kind regards,

Ivan Alejandro Pulido Tarquino, MSc

Academic Editor

PLOS ONE

Additional Editor Comments (optional):

Dear Author,

I am truly pleased to see that your article has gained greater clarity and overall quality after the review process. This is particularly important given the extremely serious and useful nature of the topic in clinical practice. It is indeed crucial to have perspectives from both sides on potential new treatments for chronic diseases like HIV. The success of treatment depends not only on the medicine itself but also on how it is presented and accepted by patients. For this reason, I encourage you to continue your research and dissemination on such important topics in the control of HIV.

Thank you

Best regards

Reviewers' comments:

Reviewer's Responses to Questions

**Comments to the Author**

1. If the authors have adequately addressed your comments raised in a previous round of review and you feel that this manuscript is now acceptable for publication, you may indicate that here to bypass the “Comments to the Author” section, enter your conflict of interest statement in the “Confidential to Editor” section, and submit your "Accept" recommendation.

Reviewer #4: All comments have been addressed

2. Is the manuscript technically sound, and do the data support the conclusions?

Reviewer #4: Yes

3. Has the statistical analysis been performed appropriately and rigorously? 

Reviewer #4: N/A

4. Have the authors made all data underlying the findings in their manuscript fully available?

Reviewer #4: No

5. Is the manuscript presented in an intelligible fashion and written in standard English?

Reviewer #4: Yes

6. Review Comments to the Author

Reviewer #4: (No Response)

7. PLOS authors have the option to publish the peer review history of their article (what does this mean?). If published, this will include your full peer review and any attached files.

Reviewer #4: No

---

## [Editor Report · Acceptance letter]

8 Aug 2024

PONE-D-22-15158R3 

PLOS ONE

Dear Dr. Petit, 

I'm pleased to inform you that your manuscript has been deemed suitable for publication in PLOS ONE. Congratulations! Your manuscript is now being handed over to our production team.

Kind regards, 

on behalf of

Dr. Ivan Alejandro Pulido Tarquino 

Academic Editor

PLOS ONE